# Uncertainty Modeling of Fatigue Crack Growth and Probabilistic Life Prediction for Welded Joints of Nuclear Stainless Steel

**DOI:** 10.3390/ma13143192

**Published:** 2020-07-17

**Authors:** Haijun Chang, Mengling Shen, Xiaohua Yang, Junxia Hou

**Affiliations:** 1School of Nuclear Science and Technology, University of South China, Hengyang 421001, China; changhaijun1@163.com; 2Nuclear Industry Engineering Research & Design Co., Ltd., Beijing 101300, China; shenmengling@126.com (M.S.); houjunxia001@163.com (J.H.)

**Keywords:** fatigue crack growth, probabilistic life prediction, welded joints of nuclear stainless steel, uncertainty, Monte Carlo simulation

## Abstract

Welded joints are widely used in the pipeline connection of nuclear power plants. Defects in these joints are an important factor leading to the failure of welded joints. It is critical to study the fatigue crack growth and life prediction methods for the welded joints with defects, to reduce their likelihood. In this paper, we present our study of the uncertainty of fatigue crack propagation and probabilistic life prediction for welded joints of nuclear stainless steel. The standard compact tension (CT) specimens were fabricated according to the American Society for Testing and Materials (ASTM) standard. Fatigue crack propagation tests with different stress ratios were performed on CT specimens, using the Mei Te Si (MTS) fatigue test system. A fatigue crack propagation rate model considering the uncertainty of material parameters, and based on the Paris formula and crack propagation experimental data, was established. A probabilistic life prediction method based on Monte Carlo simulation was developed. The fatigue crack propagation prediction result of a CT specimen was compared with the actual tested result, to verify the effectiveness of the proposed method. Finally, the method was applied to an embedded elliptical crack in welded joints of nuclear stainless steel, to predict the fatigue crack growth life and evaluate the reliability.

## 1. Introduction

As of April 2020, 47 nuclear power units have been built and put into operation in China [1]. With the increase of the service time of nuclear power units, the risk of failure of nuclear safety related components is increasing. According to the specification of “Design and construction rules for mechanical components of pressure water reactor (PWR) nuclear islands” (RCC-M), welding defects smaller than certain sizes can be accepted during the construction of nuclear power plants. In the in-service inspection of the nuclear power plant, defects beyond the acceptance range can be analyzed as cracks. The stainless-steel pipelines in the main circuit of the nuclear power plant operate under high temperatures and high pressure, and the weld joints are subject to the fluctuation of the thermal cycle load and mechanical cycle load. The fatigue cracks are easily generated and propagated [2]. The welded joint is the weak part of the welding structure. A study is necessary to investigate the fatigue crack growth behavior of the welded joint and the life prediction method of a pipeline with defects.

For the evaluation of the fatigue life, the crack initiation life can be neglected for the welded structure with defects. The fatigue life is the crack growth life of the welded structure [3]. The key points of crack growth life calculations are accurate fatigue crack growth parameters and appropriate life prediction methods. There are many studies on the fatigue crack growth of the welding joints. Rozumek et al. studied the crack path and fatigue growth in plane specimens made of S355 steel, under tension and bending with torsion, and they found that under mixed-mode condition I + II, the fatigue life increased with the loading angle value [4]. Li et al. studied the influence of micro-structure changes in different regions of 316LN welded joints on fatigue crack growth [5]. Ramy et al. studied the influence of welding heat input and welding residual stress on the stress intensity factor and fatigue crack growth behavior [6]. Lu et al. conducted fatigue crack propagation research on nuclear casting austenitic stainless steel; the results showed that the crack growth rate, *da/dN*, has a strong correlation with load ratio, *R* [2]. Tang et al. studied the fatigue crack propagation behavior of the base material, welding metal, and heat-affected zone material in the welded joints, and the results showed that the fatigue crack propagation rate of the welding sample was the highest, followed by the heat-affected zone sample and the base material sample [7]. Arora et al. determined the Paris formula constants for pipe and pipe weld materials, using compact tension (CT) specimens machined from the actual pipe–pipe weld. The fatigue crack growth life analyses of the austenitic stainless-steel pipes–pipes welds having part-through cracks on the outer surface were carried out [8].

The above research focused on the influence of the micro-structure of the welding area, load ratio (*R*), and the position of the welding joint on the fatigue crack growth rate. The material data were obtained, and the average value was used to represent the material performance of the welding parts. Due to the non-uniformity of the structure of the welding parts, the random distribution of internal defects, and the influence of some accidental factors in the processing, the mechanical properties of the welded parts have great dispersion. The fatigue analysis and structural integrity assessment were carried out on the welded parts. It is not accurate to use the average value to represent the material performance of the welded parts. To predict the fatigue crack growth life more accurately, the randomness of fatigue crack growth parameters in welding parts should be considered, and the probabilistic statistics and mechanical analysis methods should be used to solve the fatigue analysis problem. Liu found that the material coefficient was the main factor affecting the crack growth rate through the statistical analysis of various random factors in the sub-critical crack growth stage. At the same time, the parameters in the fatigue crack growth rate formula were all randomized, and the fatigue crack growth life with the given reliability was predicted [9]. Wang et al. analyzed the life prediction of the product under three failure modes, namely product sudden failure, product performance degradation, and impact failure, and established a reliability calculation model [10]. Rafiee et al. analyzed the life prediction and reliability calculation with the influence of uncertain factors [11]. Ma used the probabilistic statistical analysis method to fit the distribution type and probabilistic S–N (P–S–N) curve of the life data and used the maximum likelihood method and Kolmogorov–Smirnov (K–S) test method to estimate the parameters of the life truncated data [12].

The probabilistic life prediction method, which combines the statistical theory and probabilistic method into the fatigue life prediction model, is the most effective method to predict fatigue life [13]. In our study, the crack propagation of welded joints of nuclear stainless steel prepared by a specific welding procedure was investigated to obtain the material parameters for life prediction. Fatigue crack growth rate tests of welded joints were carried out. The distribution function of crack growth parameters was obtained with statistical theory. A probabilistic life prediction method, which combined the crack growth parameters distribution function, Paris formula, and Monte Carlo method, was developed to predict the life of welding joints with defects.

## 2. Theory of Fatigue Crack Growth

To predict the fatigue crack growth life of structures and materials accurately, studying the fatigue crack propagation behavior is necessary. Fatigue crack growth rate is one of the main parameters to describe fatigue crack propagation [14]. The Paris formula [15], which is widely used in engineering, describes the fatigue crack growth rate in a stable growth stage of crack. The Paris formula can be expressed as follows:(1)$$\frac{da}{dN}=C{\left(\Delta K\right)}^{m}$$
where *a* is the crack length, *N* is cycle number, ∆*K* is the range of the applied stress intensity factor, and *C* and *m* are parameters that are dependent on material and environmental conditions.

According to damage tolerance design, it is necessary to quickly predict the remaining life of the structural parts and make accurate and reasonable maintenance plans to ensure the safe operation of the equipment. For life prediction, the initial size of the internal defects in the components, *a*_0_, and the critical crack size, *a_c_*, determined by the fracture toughness, *K_Ic_*, of the materials or the boundary size of the components are needed.

The remaining life *N* of the components can be calculated by the fatigue crack growth rate formula with the numerical integration theory:(2)$$N=\int_{a_{o}}^{a_{c}}\frac{da}{C{\left(\Delta K\right)}^{m}}$$

For CT specimens, the stress intensity factor is calculated as follows [16]:(3)$$\Delta K=\frac{\Delta P}{B\sqrt{W}}\frac{\left(2+\beta \right)}{{\left(1-\beta \right)}^{3/2}}\left(0.886+4.64\beta -13.32\beta^{2}+14.72\beta^{3}-5.6\beta^{4}\right)$$
where ∆*P* is the range of the cyclic load; *B* is the thickness of the specimen; *W* is the width of the specimen; and *β* = *a*/*W*, *a* is the crack length.

Take the logarithm of Equation (1) to obtain the following:(4)$$\ln\frac{da}{dN}=\ln C+m\ln\Delta K$$

Equation (4) shows that the crack growth rate, *da/dN*, and the range of the applied stress intensity factor, ∆*K*, are linear equations with slope *m* and intercept ln*C* in the double logarithmic coordinate.

To calculate the fatigue crack growth life of defects in the welded joints of nuclear stainless steel, it is necessary to carry out fatigue crack growth tests on the welded joints. The fatigue crack growth rate parameters *C* and *m* of the welding materials can be obtained by utilizing the experimental data.

## 3. Fatigue Crack Growth Test of Welded Joints

### 3.1. Material and Specimen Preparation

The base material was 304L stainless steel plate, and the welding wire was ER316L wire with a diameter of 1.6 mm. The chemical composition and mechanical properties of the base material are listed in Table 1 and Table 2, respectively. Tungsten inert gas welding (TIG) technology, using argon as the protective gas, was performed for the preparation of welding materials. Welding was performed according to the RCC-M standard. The welding material, after polishing and cutting, is shown in Figure 1.

CT specimens are prepared according to the standard American Society for Testing and Materials (ASTM) E647-15 “Standard Test Method for Measurement of Fatigue Crack Growth Rates”. The geometry of the specimen is shown in Figure 2a. The thickness of the specimen was 4.5 mm, and the length of the prefabricated notch was 5 mm. Considering that the influence of the sampling direction of CT specimens in the welding material on the fatigue crack growth characteristics, 16 CT specimens with 6 different sampling directions were manufactured by using the welding material, as shown in Figure 2b. In “YZ1~3”, for example, “YZ” represents the sampling direction, and “1~3” represents 3 specimens. The serial number of specimens in the YZ direction are YZ1, YZ2, and YZ3. The prefabricated notch of the specimen is at the center of the welding seam and perpendicular to the surfaces of the welding material.

After the polishing and electrolytic erosion of the CT specimens, an optical microscope was used to observe the specimens, as shown in Figure 3. No crack or defect was found in the welding material. The welding groove was V-shaped (Figure 3a), and the welding paths were shaped like fish scales. The notch was located in the welding area and perpendicular to the surface of the welding material.

### 3.2. Fatigue Crack Propagation Test

The fatigue crack propagation test was performed on the MTS Land Mark 379.10 fatigue test machine, as shown in Figure 4. The tests were performed at room temperature. The constant amplitude loading tests with stress ratios of 0.05 and 0.79 were carried out, respectively. The experimental loads were sinusoidal alternating loads at a frequency of 10 Hz. The tests were carried out according to the ASTM E647-15 standard. A crack opening displacement meter (COD gauge) was used to obtain crack length measurements. To eliminate the effect of the notch on crack propagation, the K-drop method was used to prefabricate a crack with a length of 1 mm. The testing conditions are shown in Table 3, where 12 specimens, labeled with the numbers 1 and 2, were used for the uncertainty modeling of the fatigue crack growth rate parameters, and 4 specimens, labeled with the number 3, were used to validate the probability life prediction method.

### 3.3. Testing Result

The fatigue crack growth test was carried out for each specimen, and the change of crack length (*a*) with the number of loading cycles (*N*) was obtained, as shown in Figure 5. By utilizing the *a-N* curve, the fatigue crack propagation characteristics of materials in the welding materials can be calculated and analyzed.

## 4. Uncertainty Modeling of Fatigue Crack Growth Rate Parameters

### 4.1. Test Data Analysis

Using the *a-N* data obtained from fatigue crack growth tests and Equation (3), the crack growth rate, *da/dN*, and stress intensity factor range, ∆*K*, of each specimen can be calculated. The crack growth data of 12 specimens at the stable growth stage are shown in Figure 6. The fatigue crack growth test data of different CT specimens have great dispersion. The fatigue crack growth behavior of the welding joints of nuclear stainless steel presents a certain degree of randomness. To predict the fatigue crack growth rate more accurately, the probability statistical analysis of fatigue crack growth behavior was performed.

The material parameters ln*C* and *m* can be calculated by regression analysis, utilizing fatigue crack growth data and Equation (4). The fitting results are shown in Figure 7 and Table 4. The black circle represents the fatigue crack growth test data, and the black line represents the prediction line of crack growth rate based on the material fitting parameters in Figure 7. Some measured data deviate from the fitted prediction line, and the fitting parameters of each specimen are also dispersive. The higher deviations are due to the uncertainties from the orientation and the location of the specimens. The propagation of the cracks in the welded material, the base material, and the fusion lines have more variations, as compared to that in a more uniform material.

According to the results in Table 4, the material parameters are not constant values. If only the material parameters obtained from one specimen were used, or the average values of all material parameters were used to characterize the material characteristics, a large deviation to the fatigue life prediction and analysis of the welding joints would occur. The probability statistical analysis method was adopted to model the uncertainty of fatigue crack growth parameters of the welding joints and obtain the distribution function of material parameters.

### 4.2. Analysis of Uncertainty Modeling of Fatigue Crack Growth Parameters

The uncertainty analysis of material parameters was carried out based on the fitting data of the above specimens. The linear regression statistical analysis was performed by using 12 groups of parameters (ln*C*, *m*). The probability distribution function of parameters (ln*C*, *m*) was obtained, as shown in Figure 8.

The distribution function of fatigue crack growth parameters (ln*C*, *m*) can be described by a two-dimensional normal distribution:(5)$$p\left(\ln C,m\right)=\frac{1}{2\pi \sqrt{\left|\Sigma \right|}}\exp\left(-\frac{1}{2}{\left[\begin{array}{l} \ln C+31.2725\\ m-3.2778 \end{array}\right]}^{T}\times \Sigma^{-1}\left[\begin{array}{l} \ln C+31.2725\\ m-3.2778 \end{array}\right]\right)$$
where $\Sigma =\left[\begin{array}{c} 13.4390-2.0164\\ -2.0164\;0.3079 \end{array}\right]$.

## 5. Probability Life Prediction for Fatigue Crack in Welding Joints of Nuclear Stainless Steel

For nuclear stainless-steel welding joints, the fatigue crack growth rate has statistical dispersion. To ensure the reliability of the life prediction of the welding joints, the uncertainty of the material parameters ln*C* and *m* were considered. Monte Carlo simulation was used to predict the probability life of the welding joints by using the two-dimensional normal distribution function of the fatigue crack growth parameters (ln*C, m*).

### 5.1. Probability Life Prediction Based on Monte Carlo Simulation

The Monte Carlo (MC) method can be illustrated as follows: (1) selecting samples from the random model or specific distribution; (2) constructing a probabilistic model with similar system performance; and (3) carrying out the random test using a computer and obtaining the approximate solution of the problem [17]. The MC method has been widely used in performance evaluation [18], life prediction [19], quantum computation [20], and other fields. The MC method has three main steps: (1) Construct a reasonable probability process; (2) randomly obtain samples from the known probability distribution, to obtain the random variables that obey the known probability distribution; and (3) carry out the simulation process and use the random variable values generated from the corresponding probability distribution in each simulation, to obtain the sample of the solution.

The material parameters (ln*C*, *m*) of the fatigue crack growth rate model of the welding joints were subject to two-dimensional normal distribution. The MC method was adopted to randomly sample for generating multiple sets of (ln*C*, *m*) data. Then fatigue life can be predicted by using Equation (2). The calculation process of probability life prediction based on MC Simulation is shown in Figure 9. Firstly, the MC simulation frequency was set to *w*. Then, the variables (ln*C*, *m*) were sampled randomly, according to their probability distribution function. Then, each (ln*C*, *m*) value was respectively substituted into the life formula, to obtain corresponding life. After *w* times of simulation, *w* life prediction values were obtained. The mean life value can be calculated, and life under the given reliability can be obtained.

To verify the accuracy of this method, the probability life prediction was performed with the same stress ratio and maximum force as YZ3, YX3, XZ3, and XY3 specimens, as shown in Table 3. The MC simulation number was 10^6^, and we generated multiple sets of (ln*C*, *m*) data randomly, with Equation (5). Then, each (ln*C*, *m*) value was respectively substituted into the life formula, to obtain the corresponding life. After 10^6^ cycles of simulation, life prediction values were obtained, as shown in Figure 10. The black solid line represents the mean values of 10^6^ cycles of life prediction values, the two black dotted lines represent the 95% confidence interval, and the colored lines represent the experimental data of YZ3, YX3, XZ3, and XY3 specimens, which were all in the 95% confidence interval. The results showed that the prediction curve of fatigue crack growth behavior simulated by MC can accurately describe the real fatigue crack growth behavior of specimens. The probability life prediction method was effective.

The fatigue life distribution can be calculated based on the analysis of the fatigue crack growth curve. Assume the critical crack size (*a_c_*) is *a_c_* = 10 mm. The life prediction result is shown in Figure 11a, and the probability of failure (POF) is shown in Figure 11b.

### 5.2. Life Prediction of Welding Joint of Nuclear Stainless Steel

For the life prediction of the structure and parts, the fracture criterion of the structure should be determined. The widely used fracture criteria are material fracture threshold, *K_Ic_*, and crack length threshold, *a_c_*. The structure is considered as failed once it reaches one of the fracture criteria. The corresponding number of cycles is the fatigue life of the structure. The crack length threshold is related to the structure geometry and the fracture threshold is related to the material. The fracture threshold of 316L stainless steel (*K_Ic_* = 53.34 MPa) can be obtained from Reference [21].

For the welding joints, the location and size of defects (cracks) are random. The prediction of fatigue life is closely related to the defects. The type of defect determines the calculation of the stress intensity factor. In the paper, the defect in the welding joints was considered as the elliptical crack embedded (Figure 12), and its probability life was predicted.

The stress intensity factor of the embedded elliptical crack is as follows:(6)$$K_{I}=\frac{\sigma \sqrt{\pi a}}{\Phi }{\left[\sin^{2}\varphi +\frac{a^{2}}{c^{2}}\cos^{2}\varphi \right]}^{\frac{1}{4}},\;\Phi =\int_{0}^{\frac{\pi }{2}}{\left[1-\frac{c^{2}-a^{2}}{c^{2}}\sin^{2}\varphi \right]}^{\frac{1}{2}}d\varphi .$$

In Equation (6), *a* and *c* are the semi-minor axis and the semi-major axis of the elliptical crack, respectively, and the angle φ is shown in Figure 12.

It is difficult to obtain the ratio of the long axis to the short axis to make the prediction results more conservative in the engineering application, *a/c* = 0.4, and so the stress intensity factor, K, at *φ = π/2* was adopted. In this study, an initial crack with *a*_0_ = 0.5 mm was assumed in the welding joint. The load spectrum placed onto the components was a constant amplitude cyclic load with a stress amplitude of 300 MPa. According to the fracture threshold, *K_Ic_*, the threshold value of crack length, *a_c_* = 15.4956 mm, can be obtained. Using the probability life prediction method based on Monte Carlo sampling, 10^6^ samples were randomly selected from the distribution of the material parameters model to predict the fatigue crack growth. The prediction results are shown in Figure 13a. The failure of the structure was defined as when the crack size reaches the value *a*_c_. The corresponding life distribution is shown in Figure 13b. The predicted mean fatigue crack growth life is 1.2973 × 10^5^ cycles. The probability of failure result is shown in Figure 13c.

## 6. Conclusions

A fatigue crack growth model was established that considered the uncertainty of material parameters, and the probability life prediction method was presented. The main conclusions of this research are as follows:
(1)CT specimens manufactured from different sampling directions in the welding area were prepared to simulate the initial cracks (defects) with different orientations. The constant amplitude fatigue crack growth tests with different stress ratios were carried out to study the fatigue crack growth behavior of the welding joints comprehensively.(2)The two-dimensional normal distribution of the material parameters (ln*C*, *m*) was obtained by using the crack growth test data and Paris formula. The fatigue crack growth model, considering the uncertainty of material parameters, was established. The accuracy of the model was verified by real experimental data.(3)The Monte Carlo method was used to predict probabilistic fatigue life. The method of predicting the fatigue life of elliptical cracks embedded in the welding joints is described in detail. The method provides a quantitative criterion for maintenance and detection decision.

## Figures and Tables

**Figure 1 materials-13-03192-f001:**
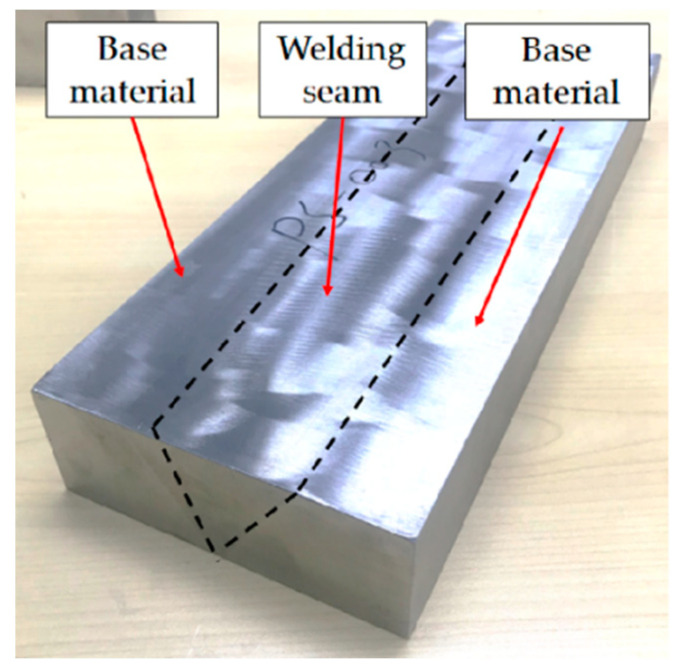
Welding materials.

**Figure 2 materials-13-03192-f002:**
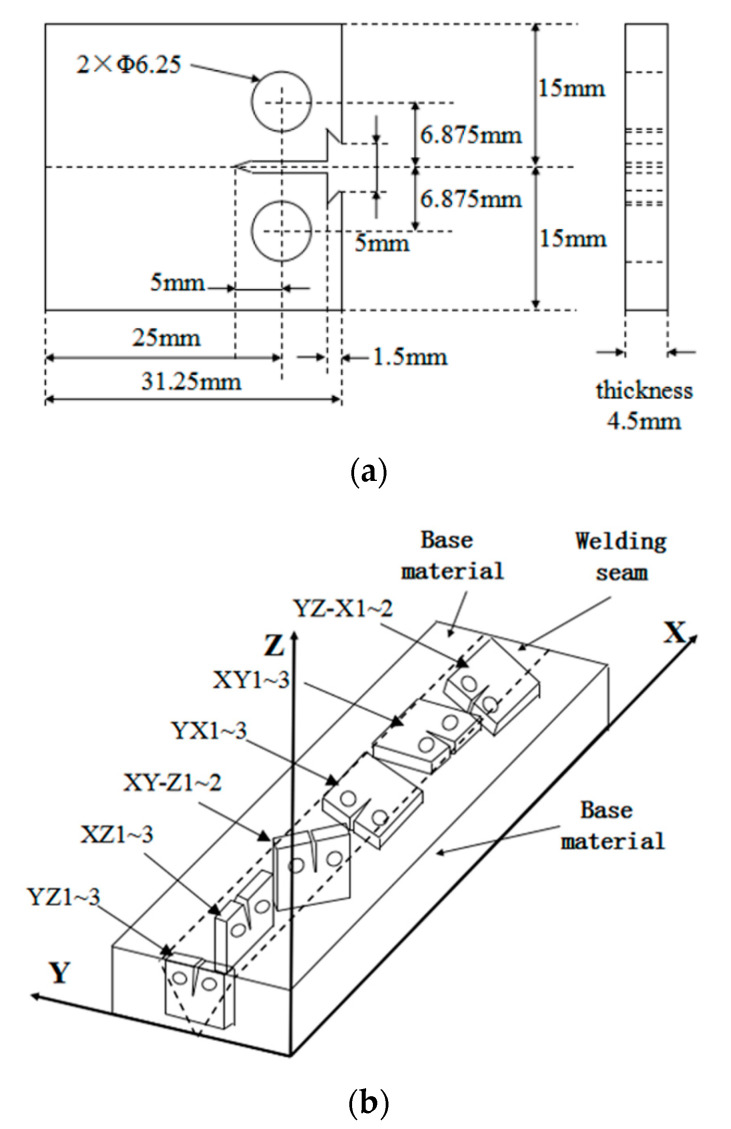
Size of specimens: (**a**) standard compact tension (CT) specimen; (**b**) 16 CT specimens manufactured by using welding material.

**Figure 3 materials-13-03192-f003:**
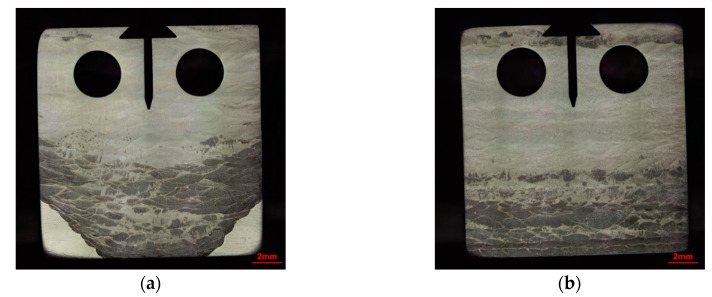
Metallographic diagram of the specimen: (**a**) XY-Z1 specimen and (**b**) XZ1 specimen.

**Figure 4 materials-13-03192-f004:**
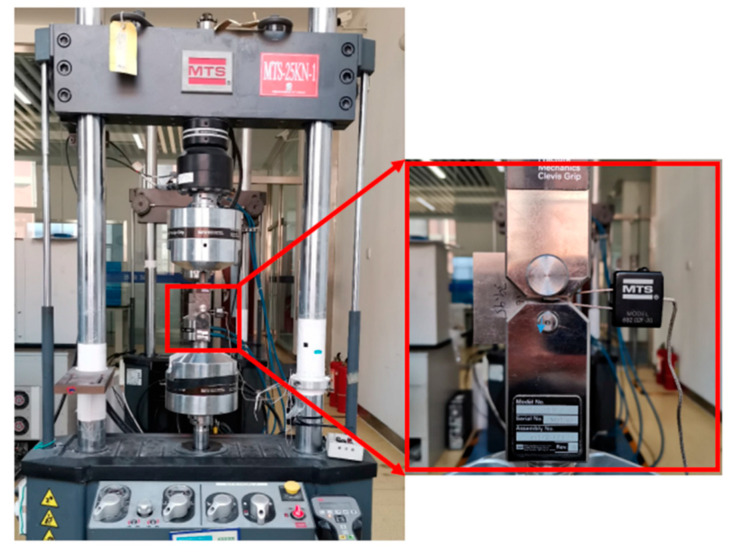
Fatigue crack growth test equipment.

**Figure 5 materials-13-03192-f005:**
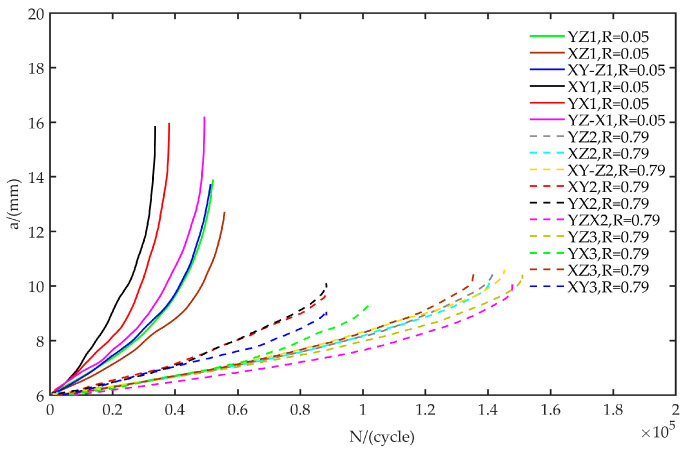
The *a-N* results of fatigue crack growth tests.

**Figure 6 materials-13-03192-f006:**
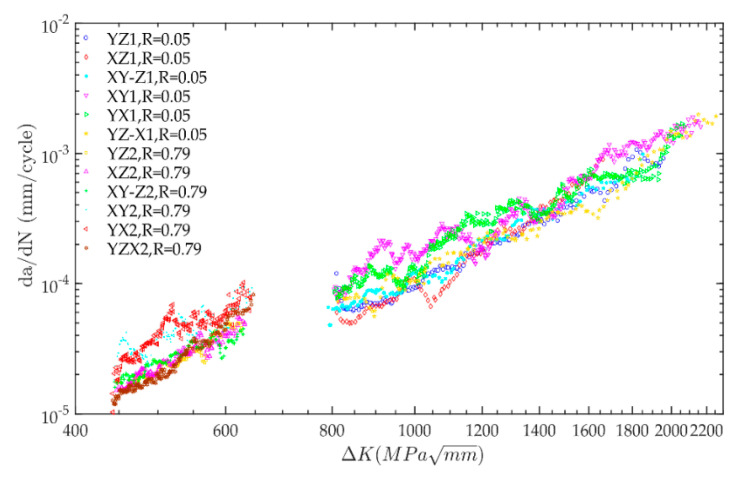
Relationship between crack growth rate, *da*/*d**N*, and stress intensity factor range, ∆K.

**Figure 7 materials-13-03192-f007:**
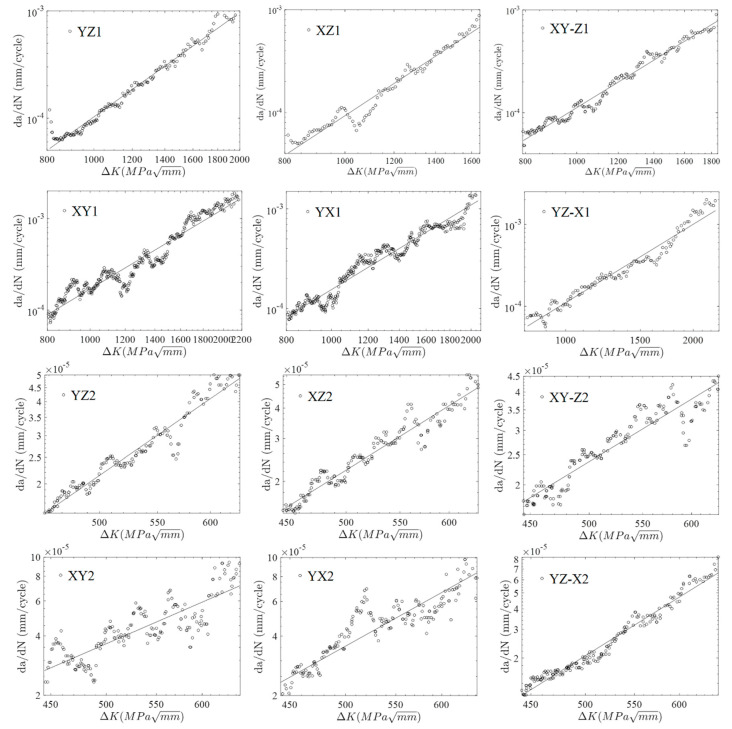
The fitting results of the fatigue crack growth rate. The corresponding code of the specimen is shown in each image.

**Figure 8 materials-13-03192-f008:**
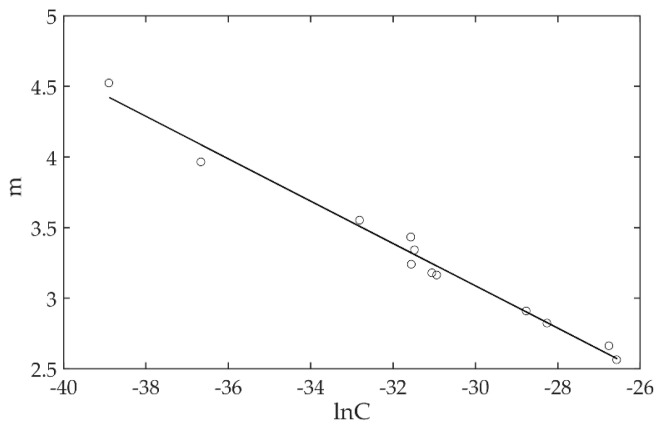
The linear regression statistical analysis of material parameters.

**Figure 9 materials-13-03192-f009:**
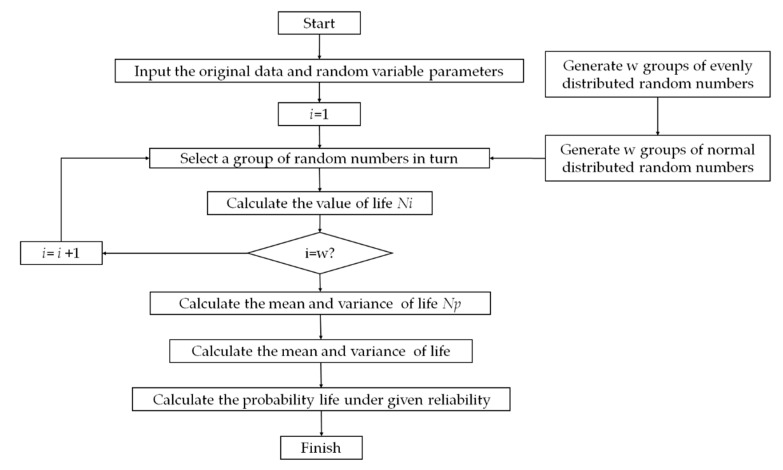
Flowchart of probabilistic life prediction by Monte Carlo simulation.

**Figure 10 materials-13-03192-f010:**
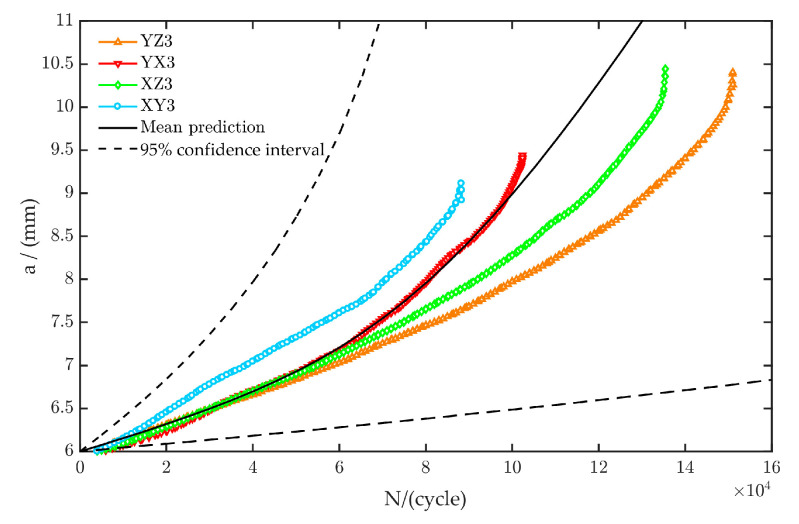
Prediction curve and actual data of fatigue crack growth behavior of CT specimens.

**Figure 11 materials-13-03192-f011:**
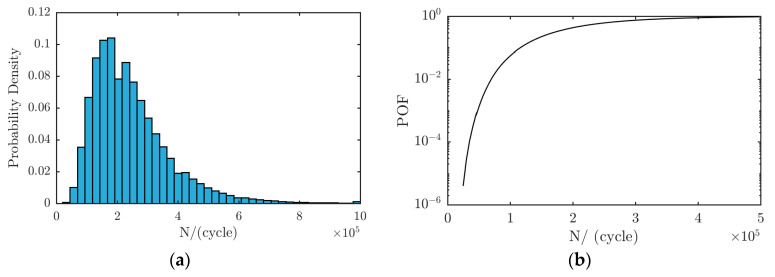
YZ3 test specimen: (**a**) fatigue life distribution and (**b**) failure probability.

**Figure 12 materials-13-03192-f012:**
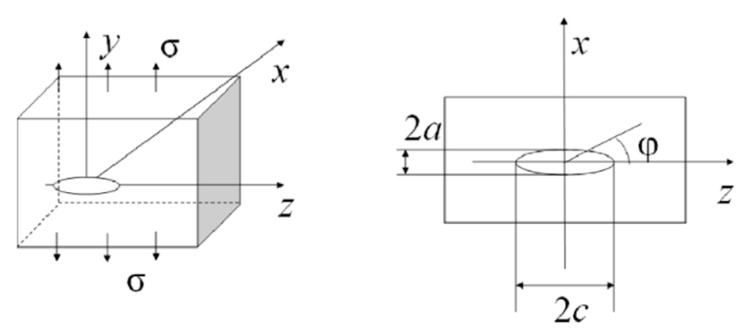
Geometric dimensions of embedded elliptical cracks.

**Figure 13 materials-13-03192-f013:**
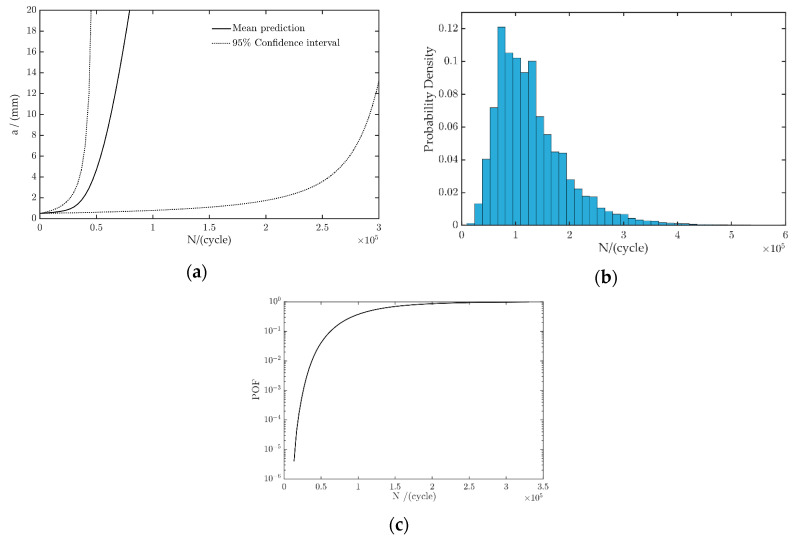
Fatigue life and reliability evaluation results of embedded elliptical cracks: (**a**) fatigue crack propagation curve, (**b**) fatigue life distribution, and (**c**) failure probability.

**Table 1 materials-13-03192-t001:** The chemical composition of welding wire material and base material (mass fraction, %).

Material	C	Si	Mn	Cr	Ni	Mo	P	S	Others
ER316L	0.025	0.42	1.91	19.10	12.58	2.57	/	/	-
304L	0.024	0.4	1.46	18.47	8.04	/	0.03	1.46	-

**Table 2 materials-13-03192-t002:** Mechanical properties of the base material.

Mechanical Properties	Values
Yield Strength (MPa)	170
Ultimate Strength (MPa)	485
Elastic Modulus (GPa)	195
Poisson’s Ratio	0.3

**Table 3 materials-13-03192-t003:** Testing setup.

Specimens	Stress Ratio, *R*	Maximum Force, *Fmax* (kN)	Frequency, *f* (Hz)
YZ1/XZ1/XY-Z1/ XY1/YX1/YZ-X1	0.05	4	10
YZ2/XZ2/XY-Z2/ XY2/YX2/YZ-X2/	0.79	10	10
YZ3/YX3/XZ3/XY3	0.79	10	10

**Table 4 materials-13-03192-t004:** Fitting results of parameters (ln*C*, *m*).

Specimen’s Code	(ln*C, m*) Fitting Result
YZ1	(−31.5528, 3.2384)
XZ1	(−36.6602, 3.9636)
XY-Z1	(−30.9338, 3.1615)
XY1	(−28.7608, 2.9073)
YX1	(−28.2575, 2.8208)
YZ-X1	(−31.0514, 3.1767)
YZ2	(−32.8073, 3.5495)
XZ2	(−31.4714, 3.3403)
XY-Z2	(−26.5650, 2.5614)
XY2	(−26.7521, 2.6604)
YX2	(−31.5637, 3.4307)
YZ-X2	(−38.8939, 4.5228)

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
