# Peer review of "Uncertainty Modeling of Fatigue Crack Growth and Probabilistic Life Prediction for Welded Joints of Nuclear Stainless Steel"

_materials, 2020, doi:10.3390/ma13143192_

Round 1

Reviewer 1 Report

Comments on the paper

  1. Page 2, line 79 - What do the authors mean and what is "... residual fatigue life ..."? Residual stress is known in science but residual fatigue life?
  2. How many specimens were tested during fatigue tests? Was the repeatability of test results?
  3. Page 5, line 140 - …cyclic loading cycles…? Improve your language.
  4. Fig. 5 – N, cycles. The same Fig. 10. In one drawing there is a parenthesis next to the unit, at other times not. Please unify.
  5. Caption under Fig. 6 - da / dn, and in the figure da / dN?
  6. It is good practice to place a table containing information on basic mechanical properties of base material as the yield strength, tensile strength, etc.
  7. Please improve the English language in the paper.
  8. It would be worthwhile to quote in the introduction also paper of: 1) Rozumek D., Marciniak Z., Lesiuk G, Correia J.A.F.O. Mixed mode I/II/III fatigue crack growth in S355 steel. Procedia Structural Integrity 2017, 5, 896-903.

Reviewer 2 Report

The article is written at a high level, is comprehensible, readable and complies with the relevant standards for its processing. The procedure for processing the experimental part was chosen correctly, it has a logical arrangement and good photo documentation. The obtained results can be used in further research.

I have no serious formal or substantive comments on this article. There are minor grammatical and stylistic errors in the article, which, however, do not absolutely reduce the level of this article.

Reviewer 3 Report

This paper studied fatigue crack growth and life prediction of welded steel joints. In total, 16 compact tension specimens were tested with two different stress ratios. A fatigue crack propagation model was established based on Paris law and experimental data. Monte-Carlo simulations were developed to predict probabilistic life. The paper is well-written; however, some changes are needed to improve its quality.

  1. References must be reviewed. There are some spelling mistakes (page 1 line 43 Lv). It is common to use for example "Li et al." in-text citations if there are more than two authors. Please check the referencing style. 
  2. The novelty of this work should be clearly presented. What is a novelty compared to articles cited in the review paper [12]?
  3. In Figure 5, only solid lines are visible. Please make sure that the dashed line is visible on the graph according to legend.
  4. In Figure 6 and 7, 12 samples are shown. The authors tested 16 samples then used 12 samples for analysis. The remaining 4 samples were used to validate the method. This should be clearly stated. It is quite confusing because of samples numbering. 
  5. Line 159, page 6. Some explanation about data scatter would be useful. Why some specimens have higher deviations than others? 
  6. In Figure 10, it is not clear which are predicted line and actual data. Can you please clarify? Have the authors included experimental data in the graph? 

Round 2

Reviewer 1 Report

  1. Page 2, line 49 - is Dariusz should be Rozumek.
